# Neuropathological Heterogeneity of Dementia Due to Combined Pathology in Aged Patients: Clinicopathological Findings in the Vallecas Alzheimer’s Reina Sofía Cohort

**DOI:** 10.3390/jcm13226755

**Published:** 2024-11-10

**Authors:** Iván Burgueño-García, María José López-Martínez, Alicia Uceda-Heras, Lucía García-Carracedo, María Ascensión Zea-Sevilla, Héctor Rodrigo-Lara, Iago Rego-García, Laura Saiz-Aúz, Paloma Ruiz-Valderrey, Francisco J. López-González, Virginia Guerra-Martín, Alberto Rábano

**Affiliations:** 1Reina Sofía Alzheimer Center, CIEN Foundation, ISCIII, 28031 Madrid, Spain; iburgueno@fundacioncien.es (I.B.-G.); mjlopez@fundacioncien.es (M.J.L.-M.); auceda@fundacioncien.es (A.U.-H.); luciagc@cajal.csic.es (L.G.-C.); mazea@fundacioncien.es (M.A.Z.-S.); irego003@ikasle.ehu.eus (I.R.-G.); lsaiz@fundacioncien.es (L.S.-A.); pruiz@fundacioncien.es (P.R.-V.); fjlopez@fundacioncien.es (F.J.L.-G.); 2Hospital Clínico Universitario Virgen de la Arrixaca, 30120 Murcia, Spain; hector.rodrigo@carm.es; 3Reina Sofía Alzheimer Center, 28031 Madrid, Spain; vguerra@clece.es

**Keywords:** dementia, Alzheimer’s disease, neuropathology, combined pathologies, longitudinal cohorts, clinicopathological correlation

## Abstract

**Background/Objectives**: Clinicopathological research in late-life dementia has focused recently on combined neurodegenerative and vascular conditions underlying the high phenotypic heterogeneity of patients. The Vallecas Alzheimer’s Reina Sofía (VARS) cohort (n > 550), and particularly the series of associated brain donations (VARSpath cohort) are presented here. The aim of this study is to contribute to research in dementia with a well-characterized cohort from a single center. **Methods**: A total of 167 patients with complete neuropathological work-ups were analyzed here. The cohort is characterized by a high female predominance (79%), advanced age at death (88 yrs.), and a high frequency of ApoE-e4 haplotype (43%). **Results**: The main neuropathological diagnosis was Alzheimer’s disease (79.6%), followed by vascular dementia (10.2%) and Lewy body dementia (6%). Overall, intermediate-to-high cerebrovascular disease was observed in 38.9%, Lewy body pathology in 57.5%, LATE (TDP-43 pathology) in 70.7%, ARTAG in 53%, and argyrophilic grain disease in 12% of the patients. More than one pathology with a clinically relevant burden of disease was present in 71.1% of the brains, and a selection of premortem neuropsychological and functional scores showed significant correlation with the number of co-pathologies identified in postmortem brains. **Conclusions**: The VARS cohort, with thorough clinical follow-up, regular blood sampling, 3-Tesla MR, and a high rate of postmortem brain donation, can provide essential multidisciplinary data in the rising age of modifying therapies and biomarkers for Alzheimer’s disease and related dementias.

## 1. Introduction

Dementia is a highly prevalent condition, most frequent among aged people, that results from various diseases affecting both cortical and subcortical brain structures. As a clinical syndrome, dementia involves disruption of memory and thinking together with increasing difficulties with daily life activities and is one of the leading causes of disability among the older population worldwide [1]. Postmortem clinicopathological studies based on longitudinal population cohorts have consistently revealed Alzheimer’s disease neuropathological change (ADNC) as the main pathological substrate of late-life dementia, followed by cerebrovascular and Lewy body pathology. Other neuropathological findings variably associated with dementia, mainly as co-pathologies, are limbic predominant age-related TDP-43 encephalopathy (LATE), argyrophilic grain disease (AGD), primary age related tauopathy (PART), aging-related tau astrogliopathy (ARTAG), and other less frequent neurodegenerative conditions (reviewed in [2]). Among the middle-aged population, clinical dementia exhibits quite a different neuropathological spectrum as a substrate, although Alzheimer’s disease (AD) is also significantly present in this age group [3].

It has been well-known for a long time that neurodegenerative pathologies, particularly those underlying dementia, appear frequently in combination. ADNC itself is a dual pathology and its hallmark histological lesions can be clearly ascribed to either extracellular amyloid-β (plaques) or intracellular tau (tangles, threads, and dystrophic neurites) pathologies [4]. It is also known that Lewy body dementia (LBD) is frequently associated with ADNC [5], and even in Creutzfeldt–Jakob disease, prion protein deposits are known to combine often with local phospho-tau pathology [6]. Similarly, cerebral amyloid angiopathy (CAA) may develop as a single pathology in elderly people, though its presence is virtually universal in AD patients [7]. Other neurodegenerative diseases, like AGD and LATE, are most frequently found in combination with other highly prevalent pathologies [8,9].

Indeed, in the last few decades, the postmortem neuropathological study of longitudinal cohorts has confirmed that combined or mixed neurodegenerative or neurodegenerative plus cerebral vascular pathologies are the norm rather than the exception in dementia patients [2]. This finding has turned critical in light of recent breakthrough advances in immune therapy for AD and of the development of biomarkers for the early detection of various neurodegenerative diseases years before their clinical manifestation [10]. Accordingly, the correlation between clinical data and postmortem neuropathological findings in patients with combined pathologies remains essential for current and future diagnostic and therapeutic approaches within the framework of precision medicine in dementia.

Since 2007, the Vallecas Alzheimer’s Reina Sofía (VARS) cohort has been receiving new participants from the Queen Sofía Alzheimer’s Center (QSAC), a monographic nursing home in the Madrid neighborhood of Vallecas, fully dedicated to the multidisciplinary follow-up of dementia patients (clinical and neuropsychological, neuroimaging, biomarkers, and brain donation) [11,12]. Over 550 patients have already been followed within the VARS cohort, while postmortem brain donation, with a complete neuropathological study, has been performed in 167 of them (the VARSpath cohort). This postmortem brain series represents one of the core research materials for all genetic, proteomic, biomarker, and neuroimaging studies developed at the CIEN Foundation (CIEN). Accordingly, all of the brains obtained from this cohort are processed through a deep neuropathological phenotyping pipeline at CIEN that includes (1) identification and histological assessment of all combined pathologies present in the brain, (2) identification of atypical forms, and (3) analysis of phenotypic variability within each identified pathology [13]. The unique research strategy represented by the VARS cohort has already yielded significant results in multidisciplinary research in dementia, particularly once the number of donated brains has reached a critical threshold [14,15,16,17,18,19,20]. A staging system for hippocampal sclerosis (HS) has been proposed based on neuropathological findings from the VARSpath in correlation with 3-Tesla MR follow-up findings [17,18]. More recently, correlation analysis between postmortem neuropathology and blood biomarkers has shown a high association between neurofibrillary pathology and serum glial fibrillary acidic protein (GFAP) that is independent from β-amyloid pathology [20].

Here, we present a full description of all pathologies, both neurodegenerative and vascular, identified in 167 brains donated to the CIEN Tissue Bank (CIEN-TB) by patients with moderate-to-severe dementia participating in the VARSpath cohort. Some selected clinical, cognitive, and functional data have been included for an additional clinicopathological analysis. In this study, we focus particularly on the prevalence, clinical significance, and observed patterns of co-pathology associated with the main neuropathological diagnoses underlying dementia. Some methodological issues concerning the assessment and analysis of co-pathologies in dementia will also be here discussed. In this article, we wish to highlight the unique opportunity that cohorts like VARS may purport for multidisciplinary research in dementia.

## 2. Materials and Methods

### 2.1. The VARS Cohort

This study includes 167 patients with moderate-to-advanced dementia that took part in the VARS cohort from 2007 to 2019 and donated their postmortem brains to the CIEN Tissue Bank (VARSpath cohort). The VARS cohort comprises all patients admitted to the Queen Sofía Foundation Alzheimer’s Centre (QFSAC) who participated in the follow-up program through informed consent by their relatives or proxies. The CIEN Foundation (CIEN) is a multidisciplinary research unit, located within the QSFAC complex, that contains clinical, laboratory, neuroimaging, and neuropathology facilities, including a brain bank with clinical autopsy facilities (CIEN-TB). Upon admission to the QSFAC, all relatives and patients are thoroughly informed about the research program carried out by the QSFAC and CIEN staff [11,12]. During the 2007–2019 period, 447 families gave informed consent (64.4% of total admitted patients) to participation in the program, and in 167 of deceased participants, postmortem brain donation was finally performed. Informed consent is scalable, so that relatives can consent separately for the use of clinical follow-up data for research, donation of blood samples to the biobank, use of genetic data of patients for population studies, performance of magnetic resonance imaging studies, and postmortem brain donation.

All patients admitted to the QSFAC bear a clinical diagnosis of Alzheimer’s or related dementia made by their reference neurologist. Upon admission and after a complete clinical work-up by the CIEN clinical team, including MR when feasible, a clinical “post-protocol” diagnosis is assigned [12]. Follow-up data and blood samples are obtained at baseline and every six months while neuroimaging studies (3-Tesla MR) are performed on a yearly basis. Thorough clinical evaluation of patients includes neurological, neuropsychological, neuropsychiatric, and functional assessment [12]. Main neurological medication of patients upon admission is summarized in Table 1. In 10% of the cases, IC for brain donation was given perimortem, even though the patients were not included in the cohort at the outset. The neuropathological data of these patients have also been included in the analysis of the VARSpath cohort.

### 2.2. Sociodemographic and Clinical Follow-Up Data

Sex, age at death, age at onset, survival time, and time spent in the nursing home (months) were registered in each case. The clinical evaluation of patients in the VARS cohort includes baseline and annual assessment of several cognitive, functional, and behavioral scales [11,12]. For clinicopathological analysis, in the present study, cognitive test results at the time of admission (baseline) and at the penultimate evaluation have been collected, to avoid the commonly observed floor or ceiling effects observed at the immediately premortem evaluation (0 to 6 months before death in the VARS program). Results from the following tests have been selected here:Semantic fluency (animal naming);Severe Mini-Mental State Examination (sMMSE);Mini Examen Cognoscitivo (MEC, a version of sMMSE adapted to Spanish and validated by Lobo et al. [21]);Neuropsychiatric Inventory Questionnaire (NPI) [22];Functional Assessment Staging Test (FAST) [23];Brief Cognitive Rating Scale (BCRS), that includes the assessment of concentration and recent and remote memory [24];Clinical Dementia Rating (CDR) [25], including global and partial memory scores (CDRm).

### 2.3. Brain Donation and Neuropathological Work-Up

Postmortem brain tissue extraction is performed at CIEN-TB on a 24 h basis. Patients of the VARS cohort may die at QSAC or at the local reference hospital. Following the CIEN-TB protocol for any brain donation, the brain bank is immediately informed of the patient’s decease by a relative and transport of the donor to the CIEN-TB autopsy facilities is rapidly arranged in coordination with the undertakers, if necessary. This quick procedure, and the proximity of many donors that die at QSAC, yields a mean postmortem interval (PMI) for the VARSpath cohort of 4 h. If the MR is available at the time of autopsy, a postmortem pre-extraction cranial 3-Tesla MR study is obtained from the donor.

Immediately after brain extraction, the fresh brain is weighed and cut into two symmetrical halves through a mid-sagittal section. The left hemi-brain is then wholly processed for neuropathological assessment and is thus fixed immediately by immersion in 4% buffered formaldehyde. As for the right hemi-brain, a transverse section through the rostral mesencephalon allows separation of infratentorial from supratentorial brain structures. The right brain hemisphere is then serially sliced following the coronal plane and starting at the level of the mammillary body, thus obtaining a series of “anterior” hemispheric slices, towards the frontal pole, and a series of “posterior” coronal slices, towards the occipital pole. The right cerebellar hemisphere is similarly sliced following the sagittal plane, while the right hemi-brainstem is left uncut. The resulting slices and tissue blocks are then wrapped, labeled, flash frozen at –50 °C, and kept for long-term storage at –80 °C (Figure 1).

After 3 to 4 weeks of fixation, the left hemi-brain is processed for neuropathological assessment. A full macroscopic examination of the whole and the sliced brain is carried out and macroscopic photographs are taken. The degree of global (0 to 3) and medial temporal lobe atrophy (0 to 3) are then assessed and registered. Thereafter, the brain and the cerebellar hemispheres are serially cut following the same plane used for the fresh right hemi-brain (sagittal), and serial transverse slices of the hemi-brainstem are additionally obtained. The size of the lateral ventricle (0 to 3) is assessed at this point. From these fixed tissue slices, a total of 25 tissue blocks are obtained from cortical and subcortical brain regions for paraffin embedding (Figure 1). This set of fixed tissue samples comprises all brain regions included in current consensus criteria for diagnosis and classification of most prevalent neuropathological conditions underlying dementia. A first set of hematoxylin and eosin (H&E)-stained sections is obtained and examined, and multiple immunohistochemical stains are thereafter performed on selected tissue blocks based on clinical, macroscopic, and H&E data. Routine primary antibodies used for neuropathological assessment of brain donations from adult patients include amyloid-β (GeneTex GTX27501 beta amyloid (1–40) antibody [BAM-10]), tau (phospho-tau (Thr212, Ser214) monoclonal antibody (AT100, Invitrogen™, Waltham, MA, USA), α-synuclein (Novocastra™ monoclonal antibody NCL-L-ASYN, clone KM51, Leica Biosystems, Nussloch, Germany), TDP-43 (10782-2-AP-150UL TDP-43 polyclonal antibody, ProteinTech, Rosemont, IL, USA), ubiquitin (ab7254 Mouse monoclonal [Ubi-1] to Ubiquitin, Abcam, Cambridge, UK), and p62 (OSS00007W-150UL SQSTM1 polyclonal antibody, Life Technologies, Carlsbad, CA, USA). Procedures and criteria applied to the histological assessment of various neurodegenerative and cerebral vascular pathologies at CIEN-TB are as follows:Alzheimer’s disease neuropathological change (ADNC): The ABC system proposed by the National Institute on Aging—Alzheimer’s Association for the classification of Alzheimer’s type pathology is employed here [26]. It is based on the separate determination of Thal amyloid stage (“A”), Braak stage for neurofibrillary pathology (“B”), and the frequency of neuritic plaques according to CERAD criteria (“C”), that are then assessed together to obtain a joint ADNC probability score (null, low, intermediate, or high). For pathological group classification, patients were classified as high-burden of ADNC if ADNC = high.Lewy body pathology (LBP): Heiko Braak’s staging system for α-synuclein (+) pathology (stages 0 to 6) is still used for all cases showing LBP, although it was originally developed for Parkinson’s disease [27]. Additionally, the recently proposed Lewy Pathology Consensus (LPC) system is applied, and this classification scheme differentiates between (i) no pathology, (ii) LBP limited to the olfactory bulb, (iii) amygdala-predominant LBP, (iv) brainstem-predominant LBP, (v) limbic, and (vi) neocortical LBP [28].Limbic-predominant age-related TDP-43 encephalopathy (LATE): Medial temporal lobe-predominant TDP-43 (+) pathology is staged according to the original definition of LATE and thus classified as (i) limited to the amygdala, (ii) involving the hippocampus, or (iii) additionally affecting the frontal cortex [29,30]. The previous staging system proposed by Josephs et al. (stages 0 to 6) is also employed [31]. The presence of hippocampal sclerosis (HS) is also registered, and a staging scheme for HS developed by our group that includes early HS stages (stages 0 to 4) is also used [17].Argyrophilic grain disease (AGD): The system published by Saito (stages 0 to 3) is used for establishing the presence and the stage of AGD [32].Aging-related tau astrogliopathy (ARTAG): A rather complex classification scheme has been proposed, as a research tool, for the registration of all possible types of regional involvement by ARTAG [33]. For the present analysis, a simple qualitative ad hoc classification has been adopted that registers presence/absence of ARTAG in the medial temporal lobe or in other hemispheric regions.Cerebrovascular disease (CVD): A total vascular score (TVS) (0 to 20) was obtained according to the scheme proposed by Deramecourt et al. [34] as the sum of partial scores corresponding to the frontal lobe (0 to 6), temporal lobe (0 to 6), basal ganglia (0 to 4), and hippocampus (0 to 4). The correspondence between the regions of interest as suggested by the original paper and the specific brain sampling performed at CIEN-TB was based on the scheme proposed by our group [14]. Patients were classified as high or low burden of CVD if their TVS was above or below the median for the whole series, respectively. Additionally, the VCING (vascular cognitive impairment neuropathology guidelines) classification system was also determined, which yields a global probability (low, intermediate, or high) for the CVD observed in a brain to contribute significantly to the cognitive decline of the patient [35]. VCING score integrates a partial score for arteriolosclerosis (0 to 4) and a partial score for CAA (0 to 4), and both scores have been included separately in the data analysis.

A selection of representative HE slides and all positive immunohistochemical stains are digitalized and kept for eventual morphometric analysis of pathological lesions and for transference of the images to researchers under request to the brain bank (BT-CIEN).

By limiting the neuropathological assessment to the left hemi-brain, the burden of some pathologies can be overestimated, particularly in the early stages. This is a limitation of diagnosis in brain banks that must be considered, especially in highly asymmetric diseases, such as frontotemporal dementias, and in the early stages of some other diseases.

For the analyses presented in Section 3 (Results), neuropathological (clinicopathological) diagnosis and pathological classification will be distinguished, when necessary, according to current guidelines, e.g., AD from ADNC, VD from CVD, and LBD from LBP. For other pathologies where no clinical diagnosis is available, a single term is used, as for CAA, LATE, HS, ARTAG, or AGD.

The following variables were defined for the assessment of combined pathology in the VARSpath cohort:CoPath (0 to 4): Number of pathologies present in a single brain among ADNC, LBP, CVD, and LATE, i.e., conditions with evidence of impact on the cognitive decline of the patient. ADNC is considered present if there is intermediate or high probability; any presence of LBP is here considered relevant; CVD is considered present if TVS ≥ median or VCING score > low probability.CoPathH (0 to 4): Number of high (H)-burden pathologies present in a single brain among ADNC, LBP, CVD, and LATE. ADNC is considered significant here if intermediate or high; high LBP is present if it qualifies as limbic or neocortical subtype on the LPC system or if Braak stage > 4; LATE is considered high burden if established HS is observed (stages 3–4 in the scheme developed by our group).CoPathT (0 to 6): total (T) number of pathologies present in a single brain, including CoPath conditions (any stage of involvement) plus AGD and/or ARTAG.

### 2.4. Genetic Analysis: ApoE Haplotype

Total DNA was isolated from peripheral blood or cerebral tissue following standard procedures. APOE genotyping (rs429358 and rs7412) was performed by real-time PCR [36]. No genetic analyses were performed in the VARSpath cohort related to autosomal dominant genetic dementias, as they were not indicated due to absence of family history.

### 2.5. Statistical Methods

Univariant analysis was performed through the description of percentages for qualitative variables and central and dispersion values for quantitative variables (mean and standard deviation for continuous and median plus quartiles 25 and 75 for ordinal variables). Bivariant analysis was based on Chi-squared or Mantel–Haenszel tests for qualitative variables and on Student’s *t*-test for analyses meeting parametric conditions, or on Mann–Whitney U test for non-parametric conditions. For more than two comparative groups, either ANOVA (parametric) or Kruskal–Wallis (non-parametric) test was performed. For comparison between two related variables, Wilcoxon signed-rank test was used. Analogously, Pearson’s test was used for correlation analysis in parametric conditions and Spearman’s as a non-parametric test. For all variables, outliers were included in the analyses if comprised within a range of data that was considered reliable. Due to the merely descriptive nature of the study, multivariant analysis was not performed, and the existence of co-variation between variables and confounders was not specifically addressed.

### 2.6. Ethical and Legal Issues

Participation of patients in the VARS cohort requires informed consent by a proxy, and postmortem brain donation to the CIEN brain bank is authorized by a relative or proxy through a separate informed consent. All ethical/legal documents of the CIEN–Tissue Bank, including informed consents, have been approved by the external ethical committee of the brain bank, i.e., the Ethical Research Committee of the Carlos III Research Institute.

## 3. Results

### 3.1. Sociodemographic and Clinical Features

The VARSpath cohort is composed very predominantly of women (79%), with a rather high age at death (mean, 87.2 years; age range, 62–105 years). Mean age at onset was 75.4 years and, correspondingly, the mean survival time was 11.9 years (range, 3–23 years) (Table 2). An age at onset below 65 years was observed in eight patients (5.4%), again with a female predominance, although with a higher participation of male patients (55.5%), and in all of them, Alzheimer’s disease was the main neuropathological diagnosis (early-onset Alzheimer’s disease). The main sociodemographic and selected follow-up data at baseline and at the penultimate evaluation, i.e., around one year before the patients’ decease, are displayed in Table 2.

The cognitive and functional data correspond to advanced dementia in most patients at baseline, with evident deterioration along around 3 years of mean follow-up in the QSAC nursing home. Differences between the baseline and penultimate values were statistically significant, except for NPI (Wilcoxon signed-rank test).

With respect to the in-house clinical diagnosis of patients (“post-protocol”), 66% of the patients were diagnosed with AD on admission, 18.4% with mixed dementia (AD plus VD), 8.2% with Lewy body dementia, 2% with VD, and 5.4% were diagnosed with other clinical conditions.

### 3.2. Main Neuropathological Diagnoses and Findings

Alzheimer’s disease (ADNC, high probability) was the main neuropathological diagnosis in around 80% of the patients, and as expected, the rest of the dementia patients received a main diagnosis either of vascular dementia (VD) (around 10%) or of Lewy body dementia (LBD) (6%) (Figure 2, Table 3). Other infrequent causes (as a main neuropathological diagnosis) of dementia were Pick’s disease, progressive supranuclear palsy, argyrophilic grain disease, tauopathy with globular glial inclusions, and hippocampal sclerosis, each one represented by one single case.

In patients with high ADNC, and particularly in early-onset cases, no neuropathological features specifically associated with genetic AD were observed, and this refers mainly to the possible presence of cotton-wool β-amyloid plaques.

As Table 3 and Table 4 show, even though Alzheimer’s disease is the highly predominant first diagnosis in this dementia cohort, a combination of pathologies, particularly of the four pathologies with evidence of clinical impact (ADNC, LBP, CVD, and LATE), is the rule. Table 3 discloses the main histological differences between the main neuropathological diagnoses, and at the same time the extensive presence of almost any pathology in any diagnostic group. However, some patterns of combination can be discerned, as, e.g., an 18.3% prevalence of amygdala-predominant LBP (Table 4), which represents most of the LBPs observed within the AD group; quite a high presence of ADNC in the LBD group, and of HS and LATE in the AD and LBD groups but not in the VD group (Table 4); or, finally, the association of AGD with VD, with a clearly different combination pattern from ARTAG (Table 4).

### 3.3. Combined Pathologies in the VARSpath Cohort

The presence of combined pathologies associated with the three main neuropathological diagnoses (AD, VD, and LBD) can be depicted at a group level, as in Figure 3, or at a single-patient level, as has been represented in Figure 4, which includes all six pathologies with a higher presence in late-life dementia. Note the almost universal presence of some degree of ADNC and cerebrovascular pathology, and the high frequency of all of the other pathologies considered.

The number of pathologies identified in each case, as registered by the quantitative variables here defined for analysis, are represented in Figure 4. Note the predominance for each variable of more than 2–3 pathologies.

The main neuropathological diagnosis and combined pathology groups, as shown, e.g., in Figure 2 (Pathological Groups 1), vary between age groups. AD is predominant in the younger age group, whereas in older age groups, there is a higher proportion of mixed and purely vascular pathologies. Additionally, the presence of non-AD, non-VD disease is higher in the 75 to 84 group for age at onset and in the 85 to 94 group for age at admission. Pure AD as the dominant pathology is absent in the group with highly advanced age (Figure 5).

### 3.4. Clinicopathological Analysis

Among patients with a main neuropathological diagnosis of AD, 69.8% were admitted to the QSFAC with a clinical diagnosis of AD, 17.6% with mixed (Alzheimer’s plus vascular) dementia, and 6.7% with Lewy body dementia. As for patients with a neuropathological diagnosis of VD, 26.7% of the cases bore a clinical diagnosis of mixed dementia, whilst none of them was diagnosed clinically with VD. As for patients finally diagnosed at postmortem with Lewy body dementia, only one case (14.3%) was suspected clinically as a synucleinopathy.

A higher survival time was significantly associated with a lower brain weight (CC = −0.421, *p* < 0.001, Spearman test) and with higher macroscopical indexes of brain atrophy. Whereas age at death was not associated with brain atrophy in the whole series, age at onset displayed a weak correlation with brain weight, global atrophy, and ventricular dilatation. Accordingly, as expected, earlier ages of onset with longer survival times were associated with more advanced brain atrophy. As for histological data related to specific pathologies, the Braak tau stage was associated with a lower age at onset and longer survival time, while the LATE stage, and particularly HS stage, showed a positive association with age at death and survival time. Braak α-synuclein stage displayed only weak association with survival time.

Survival time showed a significant correlation with all selected cognitive and functional scales (save for CDR) at the final assessment. The correlation was highest for sMMSE (CC = −455; *p* < 0.001, Spearman test). The same can be said of all macroscopic variables for atrophy, although no correlation was found with NPI. The correlation was highest between MTL atrophy and semantic fluency and sMMSE and MEC (CC > 0.4; *p* < 0.001, Spearman test).

An extensive correlation was also observed between neuropathological variables associated with Alzheimer’s pathology and some final cognitive and functional scores, with the highest level of correlation between NIA-B score and semantic fluency (CC = −0.441; *p* < 0.001), a high correlation with sMMSE and MEC, and a low or null correlation with the rest of the scores. As for other pathologies, the Braak α-synuclein stage showed a moderate negative correlation with final sMMSE and MEC, while no correlation was observed with scores for cerebrovascular pathology. Scores for TDP-43 pathology showed an extensive correlation with the final cognitive variables, save for NPI and CDR. The correlation was highest between the HS stage and final MEC or BCRS remote memory (CC > 0.3; *p* < 0.001).

As for the co-pathology variables, survival time showed a mild-to-moderate correlation with all three variables (*p* < 0.01). Additionally, all macroscopic variables related to brain atrophy showed a moderate correlation with co-pathology variables (the highest being between MTL atrophy and CopathT, CC = 0.433, *p* < 0.001). Globally, all three co-pathology variables showed low-to-moderate correlation with baseline cognitive and functional variables, and the highest correlation was observed between MEC and CopathH (CC = −401; *p* < 0.001). The same pattern was observed with the final cognitive variables, where sMMSE and MEC showed the highest correlation levels, slightly higher than the same scores obtained at baseline.

Patients bearing at least one APOE E4 allele presented higher CopatH and CopathT variables (Table 5). No significant difference was observed between the sexes for APOE E4 status.

## 4. Discussion

Here, we present the neuropathological classification of 167 brains donated in the VARS study, a cohort of institutionalized patients with dementia in Vallecas, Madrid, between 2007 and 2019 (VARSpath cohort). A selected dataset of cognitive, neuropsychiatric, and functional data of brain donors at baseline and at the penultimate evaluation before death have also been presented, together with basic sociodemographic and clinical data.

Although the age range at death was rather wide (62 to 105 years), as is usually observed in any dementia cohort, the mean age at death was high (87.3 years), with a rather long disease duration (11.9 years). This advanced age at death is consistent with a high predominance of women in the cohort (79%). The main neuropathological diagnosis was Alzheimer’s disease (high probability ADNC) in around 80% of the patients, followed by cerebrovascular dementia (10%) and Lewy body pathology (6%), and the presence of other causes of dementia (various tauopathies, mainly) as a principal diagnosis, was minimal. However, a small percentage of early onset Alzheimer’s disease (EOAD) patients were also identified in the cohort (5.4% of the total cohort, and 6% of the patients with a main neuropathological diagnosis of AD).

As expected, a high degree of comorbidity was observed (neurodegenerative and vascular co-pathologies) in the cohort. None of the brains studied in the VARSpath cohort contained just one pathology; 71% of them presented at least two pathologies with a high lesional burden, and in 18% of patients up to five different pathologies could be identified in their postmortem brain. Finally, a selected set of premortem cognitive and functional scores showed significant correlation with key variables for Alzheimer’s, Lewy body, and TDP-43 pathologies, and, as expected, with the number of co-pathologies detected in each brain.

The relative frequency of the main pathologies identified in the VARSpath cohort (ADNC, CVD, and LBP) are consistent with those observed in other population studies [2], although our cohort is limited to institutionalized people with dementia. The most prevalent diagnosis was high-probability ADNC (neuropathologically confirmed AD) in 79.6% of the patients, followed by CVD (neuropathologically confirmed VD) in 10.6%, and Lewy body pathology (neuropathologically confirmed LBD) in 6% of the subjects. However, when CVD (with a high burden of pathology) was added as a secondary diagnosis, 40.4% of the patients were found to bear high ADNC and CVD, consistent with a diagnosis of mixed dementia (Figure 2). Access to the QSFAC is controlled by the local health authorities of the Madrid Autonomous Community, based on a clinical diagnosis of Alzheimer’s disease, even though strict clinical criteria are not applied. As can be ascertained by the neuropathological findings, the cohort may be abundant in patients with mixed dementia due to small-vessel disease, while post-stroke dementia is virtually absent, as patients requiring special support for motor deficits are directed to other nursing homes within the Madrid Community. A brain donation bias must be considered in any series of brains obtained from a brain bank. Generally, this type of bias enriches the series in atypical or extreme phenotypes. However, in the VARSpath cohort, such a bias can be reasonably estimated as minimal. In a previous study comparing patients of the VARS cohort that donated their brains with those that died without brain donation, the only significant difference observed was that the former showed a higher frequency of legal guardians [37].

The VARSpath cohort illustrates in quite an explicit way the paradigm shift that has taken place in the clinicopathological interpretation of dementia towards a major role of combined pathologies [38,39,40,41]. It has been well established that highly prevalent vascular pathologies associated with ADNC, like CVD, including atherosclerosis, arteriolosclerosis, and AAC, and neurodegenerative pathologies like LBD and LATE have a measurable impact on the premortem cognitive trajectories of patients [42,43,44,45]. Moreover, the most recent interest in combined or mixed pathologies underlying dementia and other neurological conditions has focused on pathogenic aspects that could be unveiled by the observed patterns of combinations. Shared genetic risk may be a major pathogenic factor. A higher number of combined pathologies may be expected in ApoE-e4 carriers, as was also observed in the VARSpath cohort [46]. Interestingly, a wealth of data, both observational and experimental, suggests that the coexistence of multiple pathologies may not be simply passive and additive, and that synergies between them can explain the increasing phenotypical diversity observed in these patients [9,42,47,48,49,50,51,52]. These studies are unveiling new mechanisms through which various neurodegenerative pathologies might be either induced or modified by other pathologies.

Disease heterogeneity, mainly due to the co-occurrence of multiple pathologies, has been a major drawback for a precise selection of patients for clinical trials, both pharmacological and non-pharmacological. Neuropathological diagnosis still represents the gold-standard for a full and detailed diagnosis of patients, and, accordingly, for the development of biomarkers that may get closer to the real disease burden of each patient [20]. Longitudinal clinicopathological studies like the VARSpath cohort are thus essential for the development of precision medicine in Alzheimer’s and other neurodegenerative diseases [53].

## 5. Conclusions

The VARSpath cohort has generated so far relevant information on the neuropathological architecture and clinicopathological correlation in patients with dementia [14,15]. Recently, deep neuropathological phenotyping has proven particularly fruitful for establishing fine correlations between pathological traits and MR neuroimaging findings [17,18,19], with blood molecular biomarkers [20]. Soon, multiple omics studies and IA/machine learning-assisted morphometry, along with the steady increase in the number of donated brains, will represent a substantial step forward in the potential contribution of this singular cohort to multidisciplinary knowledge in late-life dementia. Such studies may support and further refine recent initiatives aimed at clinically representing the underlying heterogeneity of neuropathological findings derived principally from disease progression and the coexistence of multiple pathologies [54].

## Figures and Tables

**Figure 1 jcm-13-06755-f001:**
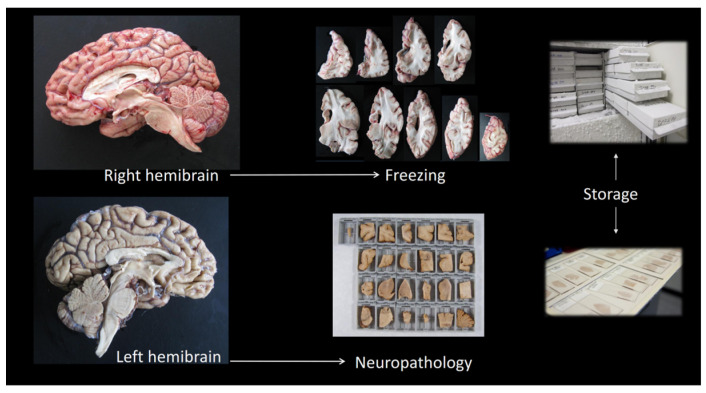
Scheme of brain tissue processing after extraction. The fresh right hemi-brain is sliced, and flash frozen for long-term storage, while the left hemi-brain is fixed in 4% formaldehyde and processed for full neuropathological examination and diagnosis.

**Figure 2 jcm-13-06755-f002:**
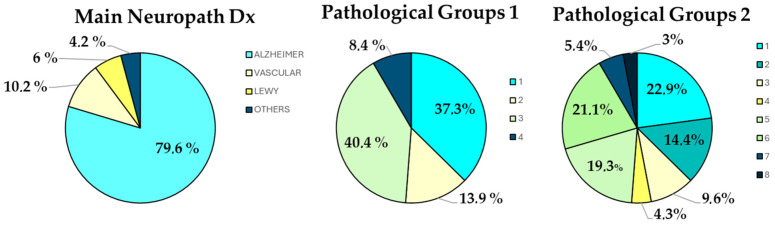
The left graph represents the distribution of main neuropathological diagnoses in the VARSpath cohort. The middle graph depicts the frequency of the four pathological groups resulting from the consideration of ADNC and CVD (high ADNC and low CVD (1); high CVD and low ADNC (2); high ADNC and CVD (3); low ADNC and CVD (4)). High ADNC = high probability of ADNC, and CVD is considered high if the Total Vascular Score (Deramecourt et al., 2012) [34] ≥ median. In the right graph, each one of the 4 pathological groups of the middle graph are split further into two groups based on the absence (odd numbers) or presence (even numbers) of any type of Lewy body pathology.

**Figure 3 jcm-13-06755-f003:**
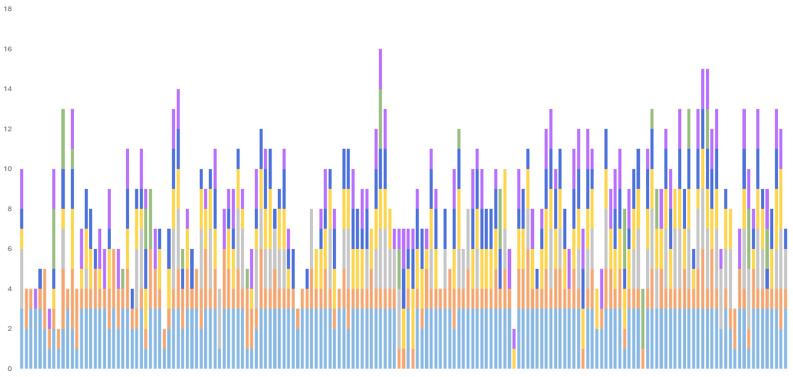
Stacked bar graph representing all patients included in the VARSpath series following a chronological order. Each bar represents a single patient, and each color segment indicates one of the most prevalent co-pathologies observed in this cohort. The height of the color segment is proportional to the intensity, stage, or probability of the represented pathology. Light blue: Alzheimer’s pathology (ADNC probability); orange: cerebrovascular pathology (VCING probability); grey: Lewy body pathology (α-synuclein Braak stage); yellow: TDP-43 pathology (LATE stage); middle blue: hippocampal sclerosis (early vs. advanced stages); green: argyrophilic grain disease (Saito stages); purple: aging-related tau astrogliopathy (limited to the MTL vs. extratemporal).

**Figure 4 jcm-13-06755-f004:**
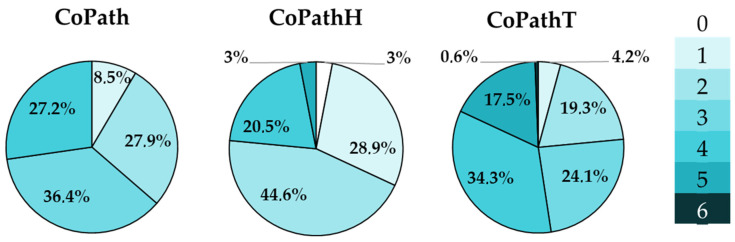
Sector graphs representing the frequencies for the different numbers of combined pathologies, including the main neuropathological diagnosis, in the VARSpath cohort, according to the definition of each co-pathology variable. See main text for the definition of each variable.

**Figure 5 jcm-13-06755-f005:**
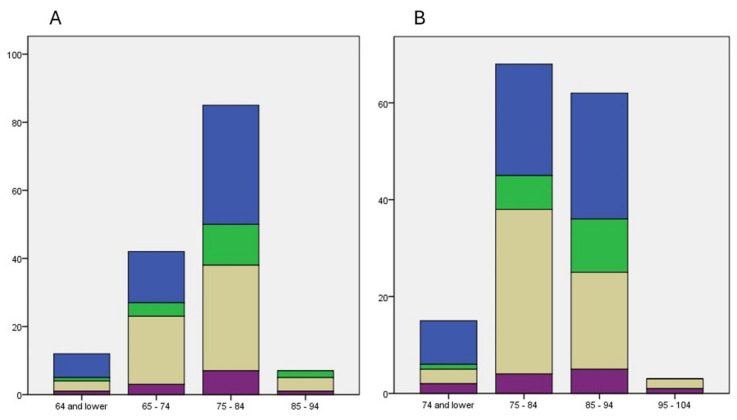
(**A**) Distribution of patients according to Pathological Group 1 (Figure 2) within age groups at onset of clinical disease. (**B**) Distribution of Pathological Group 1 within age groups defined by age at admission. Blue: high ADNC, low vascular pathology; green: low ADNC, high vascular pathology; grey: high ADNC and vascular pathology; purple: low ADNC and vascular pathology.

**Table 1 jcm-13-06755-t001:** Summary of neurological and psychiatric medication received by patients participating in the VARSpath.

Drug	% of Patients
Typical neuroleptics	24.7
Atypical neuroleptics	41.3
Antidepressants	47.3
Anxiolytics	28.2
Lithium	0
Hypnotics	26
Donepezil	47.7
Galantamine	12
Rivastigmine	25.3
Memantine	44.7

**Table 2 jcm-13-06755-t002:** Sociodemographic, clinical, genetic, and selected follow-up data of the VARSpath cohort. Follow-up data are shown for the baseline and the penultimate clinical assessment of patients, respectively.

**Sociodemographic, Clinical, and Genetic Data**		
n	167	
Sex; % (females)	79	
Age at onset, yrs.; mean (SD ^1^)	75.4 (7.3)	
Age at death, yrs.; mean (SD)	87.3 (6.5)	
Disease duration, yrs.; mean (SD)	11.9 (4.4)	
Time in nursing home, mths.; mean (SD)	52.9 (38.6)	
ApoE4 (at least one allele); %	42.8	
**Neuropsychological and Functional Scores**	**Basal**	**Final**
Semantic Fluency; mean (SD)	2.5 (4.4)	1.1 (2.6)
sMMSE; mean (SD)	14.4 (10.2)	7.9 (9.9)
MEC; mean (SD)	6.3 (6.4)	2.9 (5.1)
NPI; mean (SD)	19.3 (13.7)	21.7 (13.7)
CDR; median (IQR ^2^)	3 (2, 3)	3 (3, 3)
CDRm; median (IQR)	3 (2, 3)	3 (3, 3)
BCRS concent.; median (IQR)	6 (6, 7)	7 (6, 7)
BCRS recent; median (IQR)	6 (6, 7)	7 (6, 7)
BCRS remote; median (IQR)	6 (5, 7)	7 (6, 7)
FAST; median (IQR)	10 (7, 11)	10 (9, 13)

^1^ Standard Deviation; ^2^ Interquartile Range. BCRS concent.: Brief Cognitive Rating Scale, concentration score; BCRS recent: BCRS recent memory score; BCRS remote: BCRS remote memory score; CDR: Clinical Dementia Rating, global score; CDRm: Clinical Dementia Rating, partial memory score; FAST: Functional Assessment Staging Test; MEC: Mini Examen Cognoscitivo; sMMSE: severe Mini-Mental State Examination; NPI: Neuropsychiatric Inventory Questionnaire.

**Table 3 jcm-13-06755-t003:** Macroscopic findings and histological variables related to the main neuropathological diagnoses (Alzheimer’s disease, vascular dementia, and Lewy body disease) and additional combined pathologies (LATE, AGD, and ARTAG) in the VARSpath cohort.

**Macroscopic Findings**	
Brain weight (g.); mean (SD ^1^)	966 (137)
Global atrophy (0–3); median (IQR ^2^)	2 (2, 3)
Ventricular dilatation (0–3); median (IQR)	2 (2, 2)
Atrophy of the medial temporal lobe; median (IQR)	3 (2, 3)
**Neuropathological Variables**	
Main Neuropathological diagnosis; %	79.6 (Alzheimer’s disease) 10.2 (Vascular dementia) 6 (Lewy body dementia) 4.2 (Others)
Thal stage; % (0/1/2/3/4/5)	0.6/1.8/0.6/6.1/28/62.8
NIA A; % (0/1/2/3)	0.6/2.4/6.1/90.9
Braak tau; % (0/1/2/3/4/5/6)	1.8/5.5/6.1/7.3/38.2/41.2
NIA B; % (0/1/2/3)	0/7.3/13.3/79.4
NIA C; % (0/1/2/3)	7.2/9/15.1/68.7
ADNC; % (0/1/2/3)	2.4/6.7/13.9/77.0
Braak α-syn; % (0/1/2/3/4/5/6)	59.9/0.6/11.4/6.6/10.2/11.4
LPC; % (NP/OB/AM/BS/LI/NC)	42.5/13.7/18.3/2.6/10.5/12.4
Total vascular score; median (IQR)	8 (5, 11)
VCING score; % (low/intermediate/high)	61.1/26.9/7.8
CAA intensity (0–4); median (IQR)	2 (1, 3)
HS stage; % (0/1/2/3/4)	29.9/8.9/15.3/14.6/31.2
LATE; % (0/1/2/3)	29.3/9.8/56.1/4.9
TDP43 stage (Josephs et al. [31]); % (0/1/2/3/4/5/6)	29.3/8.5/15.9/16.5/21.3/3.7/4.9
AGD; %	12
ARTAG; %	53

^1^ Standard Deviation; ^2^ Interquartile Range. ADNC: Alzheimer’s disease neuropathological change; AGD: argyrophilic grain disease; ARTAG: aging-related tau astrogliopathy; Braak α-syn: Braak α-synuclein stage; CAA intensity: cerebral amyloid angiopathy intensity; HS stage: hippocampal sclerosis stage; LATE: limbic predominant age-related TDP-43 encephalopathy stage; LPC: Lewy pathology consensus type; NP: no LBP; OB: LBP limited to the olfactory bulb; AM: amygdala-predominant LBP; BS: brainstem-predominant LBP; LI: limbic LBP; NC: neocortical LBP; VCING score: vascular cognitive impairment neuropathology guidelines score.

**Table 4 jcm-13-06755-t004:** Comparison of key variables between main neuropathological diagnoses (Alzheimer’s disease, vascular dementia, and Lewy body dementia).

Sociodemographic, Genetic, and Clinical Data				
	AD	VD	LBD	*p*
n	133	17	10	
Sex; % (females)	81.9	64.7	60	NS
Age at onset, yrs.; mean (SD ^1^)	75.1 (7.4)	77.9(4.8)	76.5 (6.4)	NS
Age at death, yrs.; mean (SD)	87.3 (6.6)	87.3 (4.9)	88.7 (6.7)	NS
Disease duration, yrs.; mean (SD)	12.3 (4.3)	9.1 (4)	12.9 (4.9)	NS
Time in nursing home, mths.; mean (SD)	56.4 (37.8)	25.3 (31.4)	56.5 (41.6)	<0.05
ApoE4 (at least one allele); %	54.5	23.5	14.3	<0.01
**Neuropathological Variables**				
Braak tau (0–6); median (IQR ^2^)	6 (5, 6)	3 (2, 4)	3.5 (2.75, 4)	<0.001
ADNC (0–3); median (IQR)	3 (3, 3)	1 (1, 2)	2 (1, 2)	<0.001
Braak α-syn (0–6); median (IQR)	0 (0, 4)	0 (0, 0)	6 (5, 6)	<0.001
Total vascular score (0–20); median (IQR)	8 (5, 10)	11 (9, 13)	6.5 (4.25, 11.25)	=0.001
CAA intensity (0–4); median (IQR)	2 (1, 3)	1 (1, 2)	2 (1, 2)	=0.055
HS stage (0–4); median (IQR)	3 (0.5, 4)	0 (0, 2.5)	0 (0, 2)	=0.01
LATE (0–3); median (IQR)	2 (0, 2)	0 (0, 2)	1.5 (0, 2)	NS
AGD; %	7.5	41.2	20	=0.001
ARTAG; %	54.9	41.2	50	NS

^1^ Standard Deviation; ^2^ Interquartile Range. ADNC: Alzheimer’s disease neuropathological change; AGD: argyrophilic grain disease; ARTAG: aging-related tau astrogliopathy; Braak α-syn: Braak α-synuclein stage; CAA intensity: cerebral amyloid angiopathy intensity; HS stage: hippocampal sclerosis stage; LATE: limbic-predominant age-related TDP-43 encephalopathy stage.

**Table 5 jcm-13-06755-t005:** Difference in the number of co-pathologies between patients bearing [E4(+)] and not bearing [E4(−)] at least one APOE E4 allele.

Copathology Variables	E4(+)	E4(−)	*p* ^1^
CoPath	3 (0.8)	2.7 (0.9)	=0.061
CoPathH	2.1 (0.7)	1.8 (0.9)	<0.05
CoPathT	3.6 (1.1)	3.3 (1.1)	<0.05

^1^ Mann–Whitney U test.

## Data Availability

The datasets presented in this article are not readily available because the data are part of an ongoing study. Requests to access the datasets should be directed to Alberto Rábano (arabano@fundacioncien.es).

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
