# Peer review of "Neuropathological Heterogeneity of Dementia Due to Combined Pathology in Aged Patients: Clinicopathological Findings in the Vallecas Alzheimer’s Reina Sofía Cohort"

_jcm, 2024, doi:10.3390/jcm13226755_

Round 1

Reviewer 1 Report

Comments and Suggestions for Authors

Lack of Detailed Mechanistic Insights: Doesn't delve into molecular mechanisms behind interactions of co-pathologies.

Lacks discussion on how findings influence therapeutic strategies for precision medicine.

Needs clearer explanation on handling outliers and confounding variables.

Brain Donation Bias: Acknowledged but not thoroughly analyzed, possibly skewing results.

Comprehensive examination of co-pathologies in dementia, identifying significant clinicopathological correlations. Sets groundwork for further studies on dementia pathologies.

Comments on the Quality of English Language

Please check for improvements.

Reviewer 2 Report

Comments and Suggestions for Authors

The paper by Iván Burgueño-García et al. is a study characterising neuropathology in a cohort of institutionalised dementia patients in a centre of excellence. The sample size is large, and the results are of great interest in characterizing the neuropathological changes found in advanced age and moderate/severe dementia stages. There are, however, several aspects that I believe should be addressed: 
- It is not clear to me what the criteria were for the clinical diagnosis of dementia in patients admitted and, consequently, included in this analysis; given that post-mortem analysis in advanced stages (especially at advanced ages) is almost always associated (as shown by the Authors) with mixture between various types of neuropathology, knowing the clinical picture of onset could be very useful to understand not only the correlation between clinical and neuropathology but also to assess whether, based on the clinical phenotype, neuropathological aspects change in advanced stages; I, therefore, suggest that the authors report the initial clinical diagnosis of the patients included (and what criteria were used to make it) and correlate it with the neuropathology found at autopsy; 
- Relatedly, many of the aspects investigated could fit into the newly revised neuropathological criteria for the diagnosis of AD (10.1038/s41591-024-02988-7). Can the Authors use their data to determine how many of the patients initially assessed clinically as AD also meet its new definition from a neuropathological point of view?
- Did the Authors evaluate the possibility of LATE or PART neuropathology in their sample, but were the patients excluded from having a picture compatible with a diagnosis of FTD? Based on which criteria? 
- The choice of analysing only one of the cerebral hemispheres could impact the results obtained, especially in the younger age groups; indeed, there is evidence of prevalent neuropathological damage in one of the hemispheres, especially in the early stages. I advise the Authors to consider how this might have affected their analysis; 
- The genetic data is not adequately emphasised and only brief mention is made of ApoE; is there information available on PSEN1 and 2 and APP, as far as AD is concerned? Or of C9orf72, MAPT and GRN if there were also cases with suspected FTD? For example, I recommend investigating (perhaps with a dedicated paragraph) the neuropathology analysed in relation to PSEN 1 and 2 and APP mutations, to identify any peculiarities in patients carrying pathogenic mutations; furthermore, it is interesting to note that there are cases (particularly in certain geographical realities close to Spain such as Italy) of patients in whom there are double mutations, for example of PSEN1. Were there such patients among those included? In any case, how could the neuropathology differ in these patients? 
- The patients who died at a young age and who already had a diagnosis of dementia such that they required admission to a dedicated facility, what clinical diagnosis did they have? Are there neuropathological peculiarities in these? In general, it might be interesting to stratify the analyses by decade (e.g. <65 years, 65-75, etc.) and see how neuropathology changes as age progresses; 
- The Authors report that these patients also provided blood samples, but there is no mention of this aspect. This could be extremely interesting to understand whether blood biormarkers correlate with post-mortmem neuropathology; for example, were NfL, amyloid beta and pTau assessed? Recent work has shown that blood pTau is a reliable indicator of the CSF status, is this finding confirmed in your cohort? If the data is not available, I would still advise the Authors to discuss how they expect the blood data to reflect the underlying neuropathology, among which blood pTau seems to be the most promising from recent literature reviews on the correlation between blood and CSF; 
- Please add a conclusion paragraph. 

Reviewer 3 Report

Comments and Suggestions for Authors

 The authors study the neuropathological heterogeneity of dementia in aged patient in VARS cohort.
The study provide clinicopathological evidence to show that in aged dementia patients, combination of pathological changes can exist which will be useful for future AD biomarker and therapy researches.
The study investigate 167 patients with thorough clinical follow-up, blood sampling, 3-Tesla MR and a high rate of postmortem human brain samples. The data from the current study support their conclusion well, which has not been reported previous

Just one concern:

The authors should summarize the medication history of all dementia patients in a table, which can help readers to further understand the findings of the current study

Round 2

Reviewer 2 Report

Comments and Suggestions for Authors

I thank the Authors for their work. No further comments.